# Feasibility of Photodynamic Therapy for Glioblastoma with the Mitochondria-Targeted Photosensitizer Tetramethylrhodamine Methyl Ester (TMRM)

**DOI:** 10.3390/biomedicines9101453

**Published:** 2021-10-13

**Authors:** Alex Vasilev, Roba Sofi, Stuart J. Smith, Ruman Rahman, Anja G. Teschemacher, Sergey Kasparov

**Affiliations:** 1School of Life Sciences, Immanuel Kant Baltic Federal University, Universitetskaya Str., 2, 236041 Kaliningrad, Russia; otherlife@bk.ru; 2School of Physiology, Pharmacology and Neuroscience, University of Bristol, University Walk, Bristol BS8 1TD, UK or rasafi@kau.edu.sa (R.S.); anja.teschemacher@bristol.ac.uk (A.G.T.); 3Faculty of Medicine, King Abdul-Aziz University, Alehtifalat St., Jeddah 21589, Saudi Arabia; 4Children’s Brain Tumour Research Centre, Nottingham Biodiscovery Institute, School of Medicine, University of Nottingham, Nottingham NG7 2RD, UK; stuart.smith@nottingham.ac.uk (S.J.S.); ruman.rahman@nottingham.ac.uk (R.R.)

**Keywords:** glioblastoma, photodynamic therapy, photosensitizer, mitochondria

## Abstract

One of the most challenging problems in the treatment of glioblastoma (GBM) is the highly infiltrative nature of the disease. Infiltrating cells that are non-resectable are left behind after debulking surgeries and become a source of regrowth and recurrence. To prevent tumor recurrence and increase patient survival, it is necessary to cleanse the adjacent tissue from GBM infiltrates. This requires an innovative local approach. One such approach is that of photodynamic therapy (PDT) which uses specific light-sensitizing agents called photosensitizers. Here, we show that tetramethylrhodamine methyl ester (TMRM), which has been used to asses mitochondrial potential, can be used as a photosensitizer to target GBM cells. Primary patient-derived GBM cell lines were used, including those specifically isolated from the infiltrative edge. PDT with TMRM using low-intensity green light induced mitochondrial damage, an irreversible drop in mitochondrial membrane potential and led to GBM cell death. Moreover, delayed photoactivation after TMRM loading selectively killed GBM cells but not cultured rat astrocytes. The efficacy of TMRM-PDT in certain GBM cell lines may be potentiated by adenylate cyclase activator NKH477. Together, these findings identify TMRM as a prototypical mitochondrially targeted photosensitizer with beneficial features which may be suitable for preclinical and clinical translation.

## 1. Introduction

Glioblastoma (GBM) is the deadliest adult brain cancer. Among possible cellular sources of GBM are neural stem cells (NSC), oligodendrocyte progenitor cells and astrocytes [1]. GBMs exhibit a highly heterogeneous molecular makeup and are characterized by genomic instability and high tendency for infiltration. GBM exists in a variety of molecular phenotypes, including isocitrate dehydrogenase wild type, mutant type and some others [2]. Molecular heterogeneity greatly reduces chances of finding a highly potent and universally useful drug against any one specific molecular target for this type of cancer.

The global standard of care, known as the Stupp protocol [3], consists of surgical resection followed by administration of the alkylating agent temozolomide (TMZ) in combination with radio-therapy. However, the Stupp protocol only extends median survival to ~14 months from diagnosis compared to 12 months when using radio-therapy alone [3]. This treatment protocol and the resultant survival prognosis have not significantly changed for the last 15 years.

Considering that GBM does not metastatically spread around the body and that primary tumors are often reasonably well localized, it is surprising that we are still making so little progress in improving treatment outcomes. The key reason for this is that, despite maximal surgical resection, the tumor inevitably reoccurs. Interestingly, most secondary tumors arise within <2 cm of the resection edge [4]. These recurrences originate from the infiltrating GBM cells which spread from the leading edge into non-neoplastic tissue parenchyma. Today, the macroscopic boundaries of the primary tumor are usually identified by neurosurgeons using specific staining with 5-aminolevulinic acid (5ALA, trade name Gliolan^®^) which was approved by the Food and Drug Administration in 2017 for the optical detection of GBM. 5ALA is fairly selectively converted into protoporphyrin IX in cancer cells [5]. Thus, by illuminating the tumor by near-UV blue light and monitoring resultant red fluorescence, surgeons are able to detect macroscopic boundaries of GBM. However, this does not allow visualization of the microscopic infiltrations. Moreover, in many cases, even though surgeons suspect infiltrations in certain areas, they are unable to remove residual disease due to the risk of severe neurological deficits. 

Poor prognosis for patients with GBM necessitates research into alternative approaches for the treatment of this disease. One such approach is photodynamic therapy (PDT), as we have recently reviewed [6]. PDT is based on a photochemical reaction triggered by the absorption of photons of light by the molecules of a photosensitizer. Singlet oxygen and reactive oxygen species released by this reaction damage cellular macromolecules and eventually kill the cells. Even though quite a few molecules could theoretically be used as photosensitizers, only 5ALA has been extensively explored as a photosensitizer for GBM therapy, including in clinical trials [7]. The motivation for working with 5ALA is mainly the selective accumulation of fluorescent protoporphyrin IX in cancer cells which may intuitively suggest that PDT should only damage the malignant cells and not the healthy tissue. However, so far, experimental and clinical applications of 5ALA as a PDT agent have not been particularly successful [6] as 5ALA appears to be a poor photosensitizer, unable to generate sufficient amounts of free radicals to induce a powerful effect. High-power red light at approximately ~630 nm was used for its photoactivation in the published trials [8,9,10]. This contrasts with the peak absorption of protoporphyrin IX which is near 420 nm [8,9,10,11,12] This was largely motivated by the much better penetration of red light through brain tissue, but clearly, it cannot be efficient in terms of triggering the required photochemistry.

Over the past 10 years, the application of light to the brains of living animals has become a major tool in experimental neuroscience (a technology known as “optogenetics”), and a wealth of information is now available from these experiments. Hundreds of studies have used light to control cells, which are induced to express light-sensitive proteins. The wavelengths used for excitation are typically below 550 nM (blue-green-yellow). Despite the accepted notion that infra-red light (~700 nm and above) penetrates deeper into the tissue, which could be an advantage in the case of PDT, the vast experience accumulated with optogenetics unequivocally demonstrates that large quantities of light energy are damaging for a healthy brain. Moreover, light, especially the longer-wave red and infra-red light, easily releases heat which destroys brain cells (for further discussion, see [6]).

These considerations led us to investigate whether we might be more successful using another photosensitizer with a different principle of action and selectivity. 

By serendipity, we discovered that one of the dyes routinely used to image mitochondrial membrane potential (MMP), tetramethylrhodamine (TMRM), acts as an efficient photosensitizer in patient-derived primary GBM cell lines and that it is possible to achieve at least partial selectivity over non-malignant primary rat astrocytes (RA). TMRM, which is a rhodamine derivative driven into the mitochondria by their negative membrane potential has been in routine laboratory use as a research reagent but never tested as a potential therapeutic. After brief (<1 min) illumination with a green light of moderate intensity, TMRM causes the rapid and irreversible depolarization of GBM mitochondria, which ultimately leads to the apoptosis-mediated death of these cells. Here, we explored the effectiveness of TMRM as a photosensitizer for PDT (TMRM-PDT). We also attempted to increase the efficacy of TMRM-PDT using the cAMP-elevating compound NKH477 (a water-soluble analogue of forskolin) and a glycolysis inhibitor clotrimazole.

## 2. Materials and Methods

### 2.1. Primary Cultures of RA

Primary cultures of RA were prepared from the cerebral cortices, cerebellum and brainstem of Wistar rat pups (P2) as previously described [13]. Briefly, the brains of terminally anesthetized Wistar P2 pups were dissected out, crudely cross-chopped and incubated with agitation at 37 °C for 15 min in a solution containing HBSS, DNase I (0.04 mg/mL), trypsin from bovine pancreas (0.25 mg/mL) and BSA (3 mg/mL). Trypsinization was terminated by the addition of equal volumes of culture media comprised of DMEM, 10% heat-inactivated FBS, 100 U/mL penicillin, and 0.1 mg/mL streptomycin and the suspension was then centrifuged at 2000 rpm, at room temperature (RT) for 10 min. The supernatant was aspirated, and the remaining pellet was resuspended in 15 mL HBSS containing BSA (3 mg/mL) and DNase I (0.04 mg/mL) and gently triturated. After the cell debris settled, the cell suspension was filtered through a 40 μm cell strainer (BD Falcon, BD Biosciences, Franklin Lakes, NJ, USA) and cells were collected after centrifugation. Cells were seeded in a T75 flask containing the culture media (see above) and maintained at 37 °C with 5% CO_2_. Once the cultures reached confluence and 1 week later, the flasks were mildly shaken overnight to remove microglia and oligodendrocytes.

### 2.2. GBM Cell Lines

UP007 and UP029 were kindly provided by Prof. J. Pilkington (University of Portsmouth) and maintained using standard laboratory protocols in media containing 10% serum and 1% penicillin/streptomycin (0.1 mg/mL penicillin, 100 units/mL streptomycin). In some experiments, we also used primary GBM cell lines specifically derived from the infiltrative edge of surgically removed tumors as described in Smith et al. [6]. These are designated as glioblastoma invasive margin (GIN) cell lines. Their culturing conditions and handling were the same as those for UP cell lines. All of the cell lines used were of the IDH-wildtype genetic background.

### 2.3. Measurement of Cell Viability

Cytotoxicity was assessed by lactate dehydrogenase (LDH), 3-(4,5-dimethylthiazol-2-yl)-2,5-diphenyltetrazolium bromide (MTT), and PrestoBlue^TM^ (Invitrogen, Paisley, UK) assays.

#### 2.3.1. LDH Assay

LDH is an intracellular enzyme that is released from cells upon the disruption of the cell membrane or cell lysis. Thermo Scientific™ Pierce™ LDH Cytotoxicity Assay (cat no. 88954) was used to determine the toxicity of TMRM in the absence of light illumination. This is a colorimetric assay where the amount of LDH in a sample is proportional to the amount of red formazan product produced by the consumption of NADH generated by the LDH-mediated conversion of L-lactate. After adding reaction buffers to the sample culture media, color intensity in wells was measured using an Infinite^®^ 200 PRO microplate reader.

#### 2.3.2. MTT Assay

The MTT assay was used to assess the potential detrimental effects of prolonged TMRM loading on cells. RA and GBM cell lines were seeded in a 96-well culture plate at 1 × 10^4^ cells/mL in 10% FBS culture media at 37 °C in 5% CO_2_ atmosphere and allowed to attach overnight. After 24 h, the culture media were replaced with fresh media containing different concentrations of TMRM (50 nM, 100 nM, 300 nM, 800 nM) and the plates were incubated for 48 h protected from light. Tests were done in triplicates. After incubation, the MTT reagent was added into the media at a final concentration of 0.5 mg/mL and incubated for 2 h 37 °C under low light conditions. Then, the MTT reagent was removed and 100 µL DMSO in each well was added and the cells were incubated at 37 °C for 30 min to dissolve the formazan crystal precipitates. Absorbance was measured at 570 nm. Cell viability was calculated by the following formula:$$({\mathrm{Absorbance}}_{\mathrm{experimental}\;\mathrm{group}}/{\mathrm{Absorbance}}_{\mathrm{control}\;\mathrm{group}})\;\text{×}\;100\%$$

#### 2.3.3. PrestoBlue^TM^ Assay

To assess the viability of the cells after photoactivation with different illumination durations, we performed the PrestoBlue^TM^ cell viability assay (Invitrogen) on UP007, UP029 and GIN8 cell lines. This assay does not require the fixation of cells and can be performed on the same set of cells several times. Cells were seeded in 96-well plates using the technique described above. Twenty-four hours later, cell lines were loaded with TMRM (300 nM × 40 min) and photoactivated for 45 s and 90 s (1.06 mW/mm^2^). PrestoBlue assay was performed on day 2, 5 and 10 after the photoactivation of TMRM, following the manufacturer’s instructions. Briefly, 10 µL of PrestoBlue reagent was added to each well and incubated for 20 min. Fluorescence was measured at 560 nM, thus reflecting the amount of the fluorescent product of the conversion of the reagent.

### 2.4. Assessment of Basal Mitochondrial Membrane Potential (MMP) Using Potential-Driven Dye TMRM

Cells were plated in 96-well plates and allowed to attach overnight. The following day, cells were loaded with 200 nM TMRM for 1 h. Images were taken using a ZOE (Bio-Rad, Watford, UK) fluorescent cell imager. Fluorescence intensity (as an estimate for MMP) was measured using the Fiji image processing tool and compared across cell types. Image acquisition parameters were fixed across all measurements.

### 2.5. Measurements of Mitochondrial Depolarization Dynamics Caused by TMRM

TMRM decay dynamics was tested using the following protocol. The cells were plated onto glass cover slips coated with type 1 rat tail collagen at a concentration of 0.25 mg/mL to enhance the attachment of the cells. Cover slips were placed inside small corning dishes at a density of 5 × 10^4^ cells/mL. Dishes were incubated overnight in standard culture conditions. The next day, cells were loaded with 200 nM TMRM for 1 h. Before the photoactivation of TMRM, baseline images using a standard rhodamine filter block of Leica microscopes (excitation 515 nm–560 nm, emission-high pass filter 580 nm) were obtained as a sequence of six images, one every 10 s, for a total of one minute. This was followed by constant illumination with same green light for 30 s (photoactivation) followed by a series of 20 images every 10 s, for a total of 3 min. Imaging was conducted using a Leica DMIRB Inverted florescent microscope connected to a R6 Retiga digital camera and controlled by the Micromanager software. Imaging parameters such as exposure time and light intensity (1.4 mW/mm^2^, 10× objective) were fixed throughout all imaging sessions. ImageJ software was used to process the images.

### 2.6. Assessment of MMP Recovery after TMRM-PDT

In this and all other series, we used ×5 objectives (unless specifically indicated) to illuminate large areas with numerous cells to achieve a uniform biological outcome across the whole pool of cells in an individual dish. Since two different makes of microscopes were used (Zeiss and Leica) the light power density was slightly different between some datasets (1.4 and 1.06 mW/mm^2^ for Leica vs. 1.4 mW/mm^2^ for Zeiss); however, this did not qualitatively affect the outcomes.

To ensure that all cells were evenly illuminated, we used a special plating technique which ensured that they were localized in the center of the well. This was important because in the preliminary experiments, we found that the cells located at the margins of the wells and their walls do not receive sufficient quantities of light and therefore do not react to PDT.

To this end, cells were plated as 3 µL drops, containing an average of ~300–350 cells/drop at the centers of the wells in a 96-well plate. After 45 min, when cells had attached to the bottom, wells were filled with 100 µL of fresh media and incubated overnight. This plating approach was used for all experiments involving TMRM-PDT. The next day, cells were incubated with 300 nM TMRM for 40 min, and TMRM was photoactivated for 40 s using a ×5 objective (1.06 mW/mm^2^). Media in the wells were replaced and cells were returned into the incubator. Mitochondrial potential was measured 24 h later using TMRM.

### 2.7. TMRM-PDT

#### 2.7.1. Evaluating the Efficacy of TMRM as a Photosensitizer: Effect of TMRM Photoactivation on Cell Viability

Cells were plated in the centers of the well as described above and loaded with 300 nM TMRM for 45 min. Photoactivation of TMRM was carried out for 40 s (1.06 mW/mm^2^). Green light was used using standard filter blocks of Leica microscopes with a pass-band of ~520 nm–540 nm. Media were then replaced and the cells were incubated for 72 h. Fixation, nuclear staining, imaging and analysis were carried out as indicated above in this and further experiments in this section.

#### 2.7.2. Concentration-Response Test for PDT with TMRM-PDT

On day 1, cells were plated following the previously described plating protocol, and were then allowed to grow in a cell culture incubator overnight. On day 2, media were removed, and cells were loaded with TMRM (100 nM, 300 nM, 800 nM) for 40 min. The center of each well was then illuminated by green light (~530 nm–580 nm) using a LSM780 ZEISS confocal microscope with a ×5 objective at 1.4 mW/mm^2^ for 30 s. After the photoactivation, the media were exchanged and cells were incubated for 72 h. After 3 days incubation, the cells were fixed in 4% paraformaldehyde (PFA) for 15 min, washed in PBS three times and stained with 1 μg/mL DAPI for 10 min. Images were taken using a confocal microscope with objective power x5 to include all DAPI positive cells in one image for each well.

#### 2.7.3. Exposure-Dependence of TMRM-PDT

Cells were seeded as described above and loaded with TMRM (100 nM) for 40 min. Photoactivation was carried out for 30 s, 60 s or 90 s at 1.4 mW/mm^2^. Media were replaced and cells were then incubated for 72 h.

#### 2.7.4. Retention of TMRM in Mitochondria of RA and GBM Cells

Cells were seeded as described above. Cells were loaded with different concentrations of TMRM for 40 min. After 40 min, TMRM was removed and fresh media was added. After 24 h, photoactivation was performed for 30 s at 1.4 mW/mm^2^. Media were replaced with fresh media and the cells were incubated for 72 h.

#### 2.7.5. Assessment of the Effect of Green Light Alone Using Presto Blue Viability Assay

UP007, UP029 and RA were plated in the centers of the wells in a 96-well plate as described above. The next day, the cells were irradiated with green light without TMRM staining for 60 s or 120 s (1.4 mW/mm^2^). Note that the strength of this stimulus considerably exceeded all illumination regimes applied in other tests. After 3 days, cell viability was assessed with PrestoBlue assay following the manufacturer protocol.

#### 2.7.6. Incubation with NKH477, Clotrimazole and Photoactivation

Cells were seeded in 96 well-plates using the technique described above. Milder TMRM-PDT conditions were used in these experiments in order to more easily reveal any additive or synergistic effects of the combined treatment. Specifically, 200 nM TMRM was used to load the cells instead of 300 nM. A Leica EC3 florescent microscope was used to illuminate the cells with a ×5 objective lens, light dose of 1.06 mW/mm^2^ and the duration of illumination of 17 s only. To test whether NKH477 can enhance TMRM-PDT, GBM cells were pre-incubated with 10 µM NKH477 for 24 h before TMRM-PDT was conducted. The PDT outcome was tested on day 3 as previously described. We also tested whether we could potentiate the outcome of PDT with clotrimazole (a glycolysis inhibitor). Immediately after TMRM-PDT, 10 µM of the drug was also added to the wells. Cells were incubated with clotrimazole for three days before evaluating the outcome by the nuclear count as described above.

### 2.8. Assessment of Caspase Activation after PDT

To confirm that GBM cells affected by PDT undergo apoptosis, we used a genetic reporter of apoptosis CA-GFP (Caspase Activated Green fluorescent protein) as described in [14]. During apoptosis, caspase is activated via proteolytic cleavage. In CA-GFP, GFP fluorescence is completely quenched by a quenching peptide attached via the four amino acid caspase-7 cleavage motif Asp–Glu–Val–Asp. After the initiation of apoptosis, proteolytic removal of the quenching peptide by caspase-8 and caspase-9 results in restored GFP fluorescence. In order to stably express the reporter in dividing GBM cells, we generated a lentiviral vector where CA-GFP was expressed under control of the EF1α promoter, which is highly active and stable in GBM (own unpublished observation). GBM cells were loaded with TMRM (300 nM × 40 min) and photoactivated for 90 s (1.06 mW/mm^2^). The plates were kept for 10 days and the surviving cells were then fixed, stained with DAPI and imaged using a ZOE imager.

### 2.9. Lentiviral Production

The full protocol was described in our previous study [15]. Briefly, for lentiviral production, Lenti-X™293 T Cell Line (Clontech, San Francisco, CA, USA) was transfected with plasmids pNHP (7.5 μg), pHEF-VSVG (3.1 μg), pCEP4-tat (0.7 μg) and pTYF-EF1α-CA-GFP (3.9 μg). Cells were then placed in an incubator under standard cell culture conditions. Culture media were collected after approximately 30 and 48 h after transfection and stored at 4 °C. Then, the media were filtrated and centrifuged in 20% sucrose at 74,000× *g* for 2 h. The supernatant was aspirated and 25 µL PBS was added. The following day, the lentiviral vector pellet was resuspended, aliquoted and frozen at −80 °C.

### 2.10. Statistical Analysis

The data were shown as mean ± SEM; the numbers of independent experiments are indicated on the figures and in the text. Statistical analysis was performed using one-way or two-way ANOVA using Prism software version 8.00. Differences were considered statistically significant at *p* < 0.05.

## 3. Results

### 3.1. Green Light Triggers Immediate Release of TMRM from the Mitochondria

Brief exposures of TMRM-loaded (200 nM) RA and GBM cells to green light (200–300 ms every 10 s), which are required for taking images, did not affect TMRM mitochondrial localization. TMRM intensity remained stable for 30 min or more in all cell lines (data not shown).

Nuclei, under resting conditions, usually contain little TMRM but once TMRM leaves the mitochondria, it spreads into other cellular compartments including the nucleus. Therefore, results are presented as a ratio of mitochondrial/nuclear TMRM florescence intensity (Figure 1a,b).

Baseline MMP was stable before photoactivation (Figure 1a). Photoactivation of TMRM for 30 s resulted in the rapid exit of the dye from the mitochondria, most probably due to the loss of mitochondrial potential, decreasing the mitochondrial/nucleus fluorescence ratio (Figure 1a,b). Typical examples of TMRM distribution in GIN8 cell line before and after photoactivation are shown in Figure 1c,d. During this initial control period (1–30 s), it was evident that the GBM cell lines had significantly greater MMP (hyperpolarized mitochondria as reflected by the absolute intensity of TMRM staining) than normal RA, except the GIN27 cell line (Figure 2a, Appendix A).

### 3.2. GBM Cells Failed to Recover Their MMP 24 h after Photodynamic Treatment, in Contrast to Normal RA

We wanted to assess whether light-induced MMP depolarization was reversible. To this end, we re-loaded cells with TMRM 24 h after light application and measured TMRM fluorescence intensity. Note, that in this series, we did not calculate the ratio as in the previous section because we were interested in the absolute intensity values rather than the dynamics of the process. As shown in Figure 2b, the mitochondria remained depolarized in all human GBM lines, with the exception of GIN27. Importantly, mitochondria in RA fully recovered their membrane potential, evident by normal TMRM loading after 24 h.

### 3.3. Photodynamic Activation of TMRM Decreased GBM Survival

To study the effect of TMRM-PDT on GBM cell survival, cells were loaded with TMRM (300 nM) and exposed to green light. Three days later the density of the DAPI-positive nuclei was strongly reduced in wells subjected to TMRM-PDT compared to controls (see Figure 3). Curiously, this was also seen for the GIN27 line which seemed to be less affected by PDT in previous experiments (Figure 1 and Figure 2).

### 3.4. Milder Treatment Regimes Help Achieve a Preferential Effect on GBM Cell Lines

Given that GBM mitochondria were generally hyperpolarized, we sought to determine whether this could lead to a preferential effect of PDT on GBM, using milder treatment. This was only tested on UP lines for operational reasons.

Thus, 30 s, 60 s, or 90 s (1.4 mW/mm^2^) and a lower concentration of TMRM (100 nM) were evaluated. As shown in Figure 4, the number of DAPI-positive nuclei 3 days after photoactivation was significantly reduced in GBM lines but not in RA following milder treatment regimens.

### 3.5. Effect of TMRM Is Concentration Dependent

In order to better demonstrate the dependence of the TMRM effect on its concentration, we applied different concentrations while using a slightly shortened duration of illumination (30 s) in order to preserve at least some cells in stimulated wells—which was important for counting purposes. The number of DAPI-positive nuclei was not affected with TMRM loading at 100 nM while 800 nM significantly affected all tested cell types (Figure 5). In fact, the effect on both GBM cell lines was much greater than on RA (*p* < 0.001 in both cases).

### 3.6. TMRM-PDT with Preloading of the Dye

Mitochondrial membranes of GBM cells are generally more polarized relative to normal cells such as RA (Figure 2a). Thus, TMRM should theoretically accumulate and remain in the mitochondria longer in GBM cells, making them more vulnerable to TMRM-PDT. Indeed, TMRM-PDT applied to cells preloaded with TMRM for 40 min, 24 h before application of the light, had a preferential effect on the GBM lines compared to RA (Figure 6, Appendix A).

### 3.7. TMRM or Green Light Is Not Toxic to GBM Cells or RA

We controlled for the detrimental effects of TMRM without light application (dark toxicity) for RA and all GBM lines. Cells were loaded with different concentrations of TMRM for 40 min and later tested with the LDH-assay. No cytotoxicity of TMRM was observed with all five concentrations used (Figure 7).

Moreover, we increased the time of incubation with TMRM up to 48 h without light application (dark toxicity) for the RA and two GBM lines (UP007 and UP029). Incubation for 48 h with TMRM at concentrations between 50 nM and 300 nM had no significant effect on the proliferation of either of the two GBM lines or the RA, while 800 nM had some effect on UP007 and RA (Figure 8a). For the control of the light effect, illumination was carried out for 60 s and 120 s (1.4 mW/mm^2^) which are harsher conditions than in any other series, but this had no effect on either type of cells (Figure 8b).

### 3.8. PDT Effect on Viability of GBM Cells Is Long Lasting

Even though PDT resulted in a rapid loss of many GBM cells, a few were still visible in the wells after 10 days of culturing post PDT. We therefore compared their viability using the PrestoBlue assay which can be performed on the same batch of cells longitudinally and reports the metabolic health status of the cells. As shown in Figure 9a,b, UP007 and UP029 cell lines after 2, 5 and 10 days had a severely compromised metabolic status. We assumed that the main mechanism of cell death after TMRM-PDT is apoptosis. In order to visualize this process, we used a genetically encoded Caspase sensor CA-GFP as listed in the Methods. Lentiviral transduction of the cell lines was performed to generate stably expressing clones. Ten days after TMRM-PDT for 60 s (1.06 mW/mm^2^), photoactivation green fluorescence was clearly visible in most surviving cells in both cell lines, but not in the untreated controls (Appendix A).

### 3.9. NKH477—But Not Clotrimazole—Enhances the Effect of PDT in Several GBM Cell Lines

Perfect illumination of target areas and equal TMRM loading of the cells is much harder to achieve in a surgical theatre than in the research laboratory. Therefore, it is always desirable to develop additional strategies to potentiate the effect of PDT. Elevating cAMP levels in GBM cells has been implicated in multiple studies as a mechanism which could counter GBM aggressiveness and improve survival [16]. GBM cells were therefore incubated with 10 µM of adenylate cyclase activator NKH477 for 24 h before a sub-optimal TMRM-PDT regime was applied. We found that the dual targeting of GBM cells with a sub-lethal dose of NKH477 and low-intensity PDT (20 s illumination) resulted in the decreased viability of some GBM cell lines. This difference was statistically significant compared to either one of the treatments alone and to negative controls (Figure 10). Glycolysis inhibition with clotrimazole was also attempted to further drain glycolytic energy source after TMRM-PDT, which theoretically compromises mitochondrial functions, but this dual treatment protocol did not result in any significant additive effect (Figure 10).

## 4. Discussion

We here re-evaluated the potential of PDT to therapeutically target the infiltration of residual GBM disease within the brain parenchyma adjacent to the area of surgical resection, with the aim of reducing the number and viability of surviving tumor cells, and thus, improve patient prognosis. Controversies surround the use of 5ALA as a photosensitizer for PDT. First, the main peak of its excitation is 405 nm–420 nm, but this wavelength essentially does not spread in brain tissue. Furthermore, the wavelengths > 600 nm which were attempted for PDT with 5ALA may penetrate deeper into the tissues, but they are not efficient for 5ALA excitation. Second, 5ALA has a low ROS yield, meaning that it generates a small amount of free radicals upon photoactivation, and this becomes a major issue when long wavelength light is used [6].

In this study, we demonstrated the application of the MMP-driven dye TMRM as a photosensitizer for PDT targeting GBM cells. TMRM is a member of the rhodamine family and is commonly used to measure MMP in cells [17]. Changes of MMP directly correlate with changes in TMRM florescence intensity [18]. TMRM is highly mobile and instantly leaves the mitochondria if not retained by the MMP. It has also been reported to have minimal non-specific (non-mitochondrial) accumulation and interference with mitochondrial respiration compared to other commonly used rhodamine derivatives [19]. Moreover, TMRM has peak excitation/emission wavelengths of 548 nm/573 nm, respectively. Thus, TMRM can be effectively excited by a light of green/yellow spectrum. In comparison with the 405 nm–420 nm peak for 5ALA, this must increase the efficiency of PDT due to the better tissue penetration of these wavelengths.

We found that TMRM-PDT is effective in compromising the viability/survival of the GBM cells, as shown in Figure 3. Unfortunately, the effect was also seen on normal RA. However, by carefully tweaking the protocol in terms of TMRM loading concentration and illumination times, we were able to achieve significant cytotoxic selectivity to GBM cells over RA (Figure 4). Possibly, using lower concentrations of TMRM in vivo might help to achieve a selective suppressant effect on GBM infiltrating cells. The ability of GBM mitochondria to accumulate more TMRM and better retain it matches with the generally known tendency of tumor cells to have hyperpolarized mitochondria [20,21]. We believe that greater selectivity can be achieved with the protocol where photoactivation takes place after the cells are allowed to dissipate the TMRM initially loaded into them. In our experiment, this was demonstrated using 48 h delay (Figure 6). We assume that the selective suppression of the GBM cells was due to the retention of the dye in their mitochondria because of the hyperpolarized MMP. This could and should make the clearance of the dye out of normal cells more efficient than in tumor cells. From a clinical perspective, this protocol would require a preloading of the tissue with TMRM prior to light application which should follow after a delay. One needs to take into account that, unless the molecules are actively retained in brain cells in vivo, they are going to be very quickly washed away into the general circulation. Thus, in the living brain, it might only take 1–2 h for the healthy cells to release TMRM while the GBM cell could still have it concentrated in their mitochondria.

In this work, we did not study the potential impact of PDT on neurons, but this is a particularly difficult task in vitro. Essentially all experiments on cultured rodent neurons employ cells from embryos which have a completely different metabolic profile compared to the mature neurons in the brain in vivo. We do not believe that such cultures would be a suitable model for this type of work. Instead, it may be better to test whether TMRM strongly affects neurons in rodent studies in vivo, as we hope to do in future studies.

We were also able to show that TMRM is not toxic without light illumination at concentrations up to 3.2 µM, which is consistent with previous data [18]. Further, it requires comparatively low power to elicit specific cytotoxicity which decreases the chances for light-related tissue damage (see [6] for further discussion).

We attempted to enhance the efficacy of TMRM-PDT by employing a sub-lethal PDT regimen combined with NKH477 or clotrimazole. Clotrimazole is an inhibitor of phosphofructokinase, one of the key enzymes in glycolysis [22]. It has previously been tested on GBM cells and has resulted in the blockade of the cell cycle and consequently cell death. We reasoned that TMRM-PDT would perturb mitochondrial energy production and the addition of a glycolysis inhibitor would further compromise cellular energy sources and induce cell death. Unfortunately, clotrimazole at the low concentration we used (10 µM) was enough to affect RA. Moreover, no additive effect was detected on GBM cells (Figure 10). However, a combination with NKH477 has shown some promising results. NKH477 is a water-soluble forskolin hydrochloride derivative that can directly stimulate adenylate cyclase, the generator of cAMP [23]. cAMP elevation is known to cause detrimental effects on GBM cells [24,25,26]. Moreover, it has been documented that an elevated cAMP level in GBM leads to hyperpolarization of mitochondria [25] and would theoretically lead to the greater accumulation of TMRM in GBM cells and enhance the PDT effect. Interestingly, NHK477 significantly potentiated the effect of TMRM-PDT only for UP007 and GIN27 GBM cells (Appendix A). A trend was noted with other cell lines, but differences were not statistically significant (Figure 10). Heterogenous responses of GBM lines to this approach are not surprising because these tumors are characterized by a non-uniform molecular makeup and biological behavior. A hypothetical explanation for the effect of NKH477 is that it changes gene expression in GBM cells and via yet poorly defined mechanisms leads to the further hyperpolarization of their mitochondria, as illustrated by Appendix A.

## 5. Conclusions

There is an urgent need for more innovative and less invasive therapeutic modalities to treat GBM. The major source for GBM recurrences is that of the infiltrating GBM cells that are left after surgical resection for which local therapy modalities could be used to efficaciously target residual disease cells.

Its proposed topical application of TMRM may raise some of the usual concerns, such as liver or kidney toxicity, but might be less problematic because the overall dose delivered into the brain will be fairly small, especially because only a periphery of the postoperative cavity needs to be impregnated with it. As a first step, one could start by testing the consequences of injecting TMRM into the brain of experimental animals and then checking for signs of pathology and inflammation.

We envision a treatment protocol for GBM where, following surgery, the walls of the cavity and ~2 cm of the surrounding parenchyma are infiltrated with TMRM, and after a delay to allows healthy brain cells to expel the photosensitizer, light is directly delivered into the brain parenchyma where disease infiltrations are suspected.

Light delivery systems for this type of surgery are already being developed [27] and our early findings encourage continued enthusiasm in this research area.

## Figures and Tables

**Figure 1 biomedicines-09-01453-f001:**
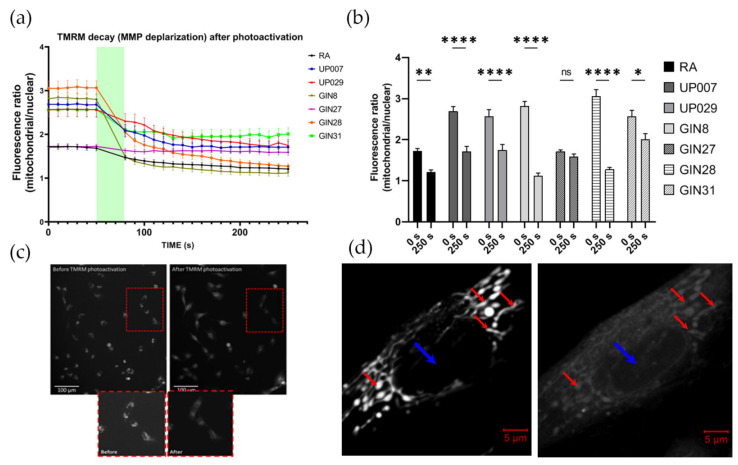
Light activation of TMRM triggers an immediate mitochondrial depolarization and results in dye re-distribution. (**a**) Mitochondria/nucleus fluorescence ratio dropped by 50% within 60 s and further decayed for the next 4–7 min. n: 24 cells from 4 independent exp. (**b**) Mitochondria/nucleus fluorescence ratios decreased after 250 s following light activation. (**c**) Representative images showing loss of TMRM from mitochondria over time and appearance of the dye in the nucleus. Red squares point to an enlarged area demonstrating a few cells at higher magnification (below). Note the redistribution of fluorescence from the bright clusters of mitochondria and its exit from the cells. The brightness of the images on the inset (lower images) is increased to facilitate viewing. For measurements, only raw images were used. (**d**) An example of a typical response to TMRM-PDT of a GBM GIN8 cell. Initially (left image), TMRM is localized exclusively to the mitochondria (red arrows) while the nucleus (blue arrow) is almost completely devoid of the staining. After photoactivation (right image), TMRM leaves the mitochondria and the contrast between mitochondria and nucleus is lost—hence the ratio mitochondria/nucleus drops). For clarification, images were taken at high magnification using LSM780 confocal microscope. (*) *p* value < 0.005, (**) *p* value < 0.001, (****) *p* value < 0.0001. ns—not significant.

**Figure 2 biomedicines-09-01453-f002:**
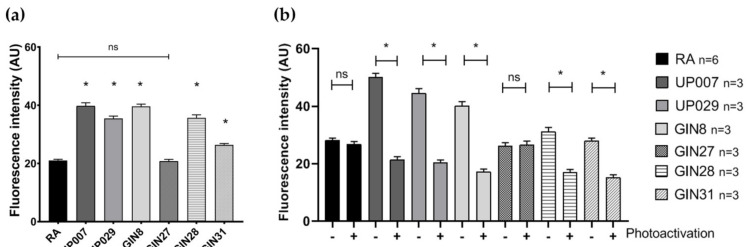
Mitochondrial membrane potential in RA and GBM cells, as reflected by TMRM florescence intensity. (**a**) Baseline MMP. GBM cells, except for GIN27, have greater basal MMP than normal RA. (**b**) Recovery of MMP 24 h after photoactivation. Normal RA and GIN27 GBM cells successfully recovered their normal MMP 24 h after photoactivation. Loss of MMP in other GBM cells persisted after 24 h. N: represents the number of independent experiments. 15+ cells were examined in each experiment. (*) *p* value < 0.0001, (ns) *p* value > 0.9999 (not significant).

**Figure 3 biomedicines-09-01453-f003:**
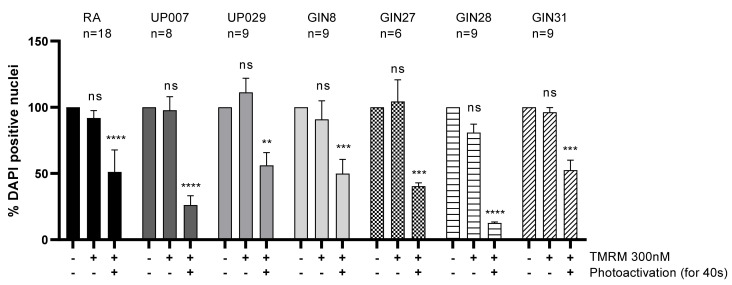
Three days after TMRM-PDT (300 nM; 40 s; 1.06 mW/mm^2^), density of cells was reduced. N is the total number of repeats from 7 independent experiments for RA, 2 for GIN27 and 3 for the other GBM cells. (**) *p* value < 0.003, (***) *p* value < 0.001, (****) *p* value < 0.0001. ns—not significant.

**Figure 4 biomedicines-09-01453-f004:**
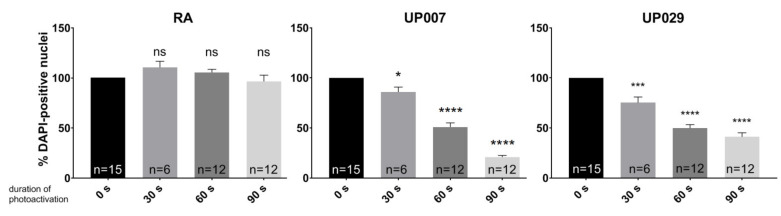
Dependence of TMRM-PDT (100 nM; 0 s, 30 s, 60 s, 90 s; 1.4 mW/mm^2^)) on the duration of illumination. N is the total number of repeats from 4 independent experiments. (*) *p* value < 0.05, (***) *p* value < 0.001, (****) *p* value < 0.0001. ns—not significant.

**Figure 5 biomedicines-09-01453-f005:**
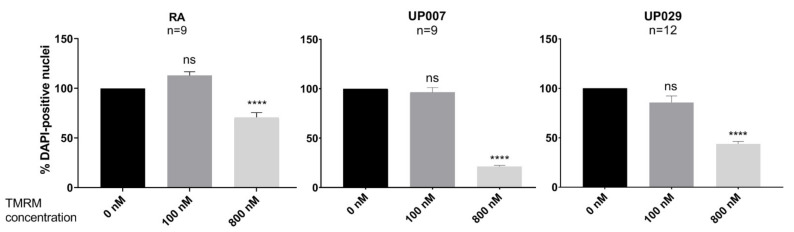
The dependence of TMRM-PDT (0 nM, 100 nM and 800 nM for 30 s; 1.4 mW/mm^2^) on TMRM concentration. N is the total number of datapoints from 4 independent experiments. (****) *p* value < 0.001. Differences between the effect of 800 nM on RA (relative decrease in cell density) and either of the GBM cell lines were highly significant (*p* < 0.01 in either case). ns—not significant.

**Figure 6 biomedicines-09-01453-f006:**
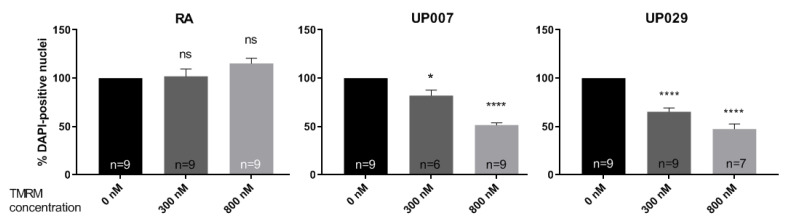
Delayed photoactivation of TMRM enables selective effect on GBM cells. Cells were loaded with TMRM, 300 nM or 800 nM which was then removed from the media. Twenty-four hours later, light was applied for 30 s (1.4 mW/mm2). When counted 3 days later, the number of remaining GBM but not RA was strongly decreased. This specificity was achieved by preferential retention of TMRM in GBM cells. N is the total number of datapoints from 3 independent experiments. (*) *p* value < 0.05, (****) *p* value < 0.0001. ns—not significant.

**Figure 7 biomedicines-09-01453-f007:**
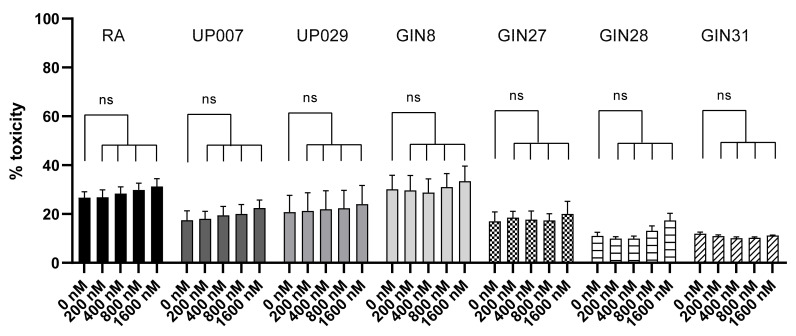
Control tests for TMRM dark toxicity. TMRM dark toxicity in the absence of illumination assessed using LDH assay. No statistically significant difference in toxicity between any of the 5 concentrations used in this experiment compared to the controls. N = 9: is the total number of datapoints for each condition from 3 independent experiments. ns—not significant.

**Figure 8 biomedicines-09-01453-f008:**
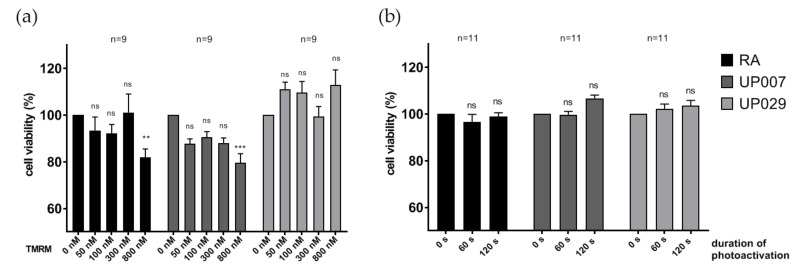
Further control experiments to assess the dark toxicity of TMRM and the effect of green light. (**a**) Prolonged incubation with TMRM for 48 h with no photoactivation had a minimal effect on GBM or RA cell density as measured using MTT assay. (**b**) When applied to unloaded cells, green light 1.4 mW/mm^2^) on its own, had no effect on the survival of GBM cell lines or RA (PrestoBlue assay). N: is the total number of datapoints for each condition from 3 independent experiments (6 for RA in panel A.). (**) *p* < 0.01, (***) *p* < 0.001. ns—not significant.

**Figure 9 biomedicines-09-01453-f009:**
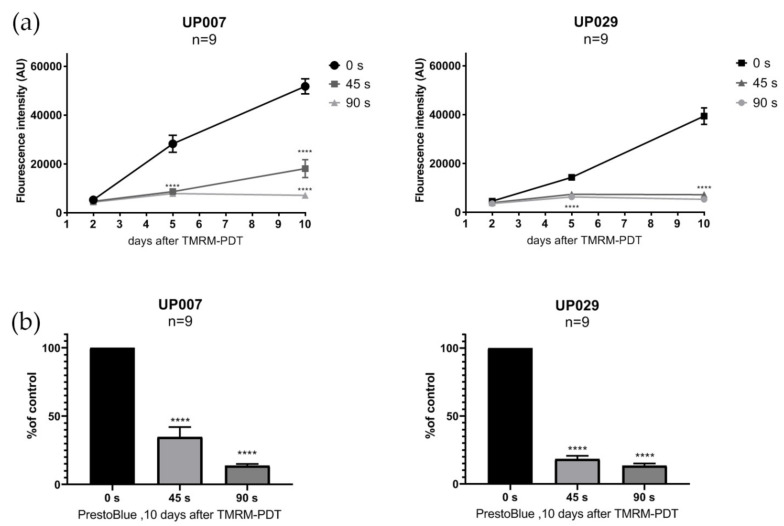
Lasting effects of TMRM-PDT. (**a**) Single episode of TMRM-PDT permanently affected the viability of GBM cell lines (PrestoBlue assay). GBM lines were photoactivated for 0 s, 45 s or 90 s after being loaded with 300 nM TMRM. N: is the total number of datapoints from 3 independent experiments. (****) *p* value < 0.0001. Data are given as the mean ± SEM. (**b**) Effect of varying light exposure on cell viability assessed 10 days after TMRM-PDT, Presto Blue assay. (****) *p* value < 0.0001.

**Figure 10 biomedicines-09-01453-f010:**
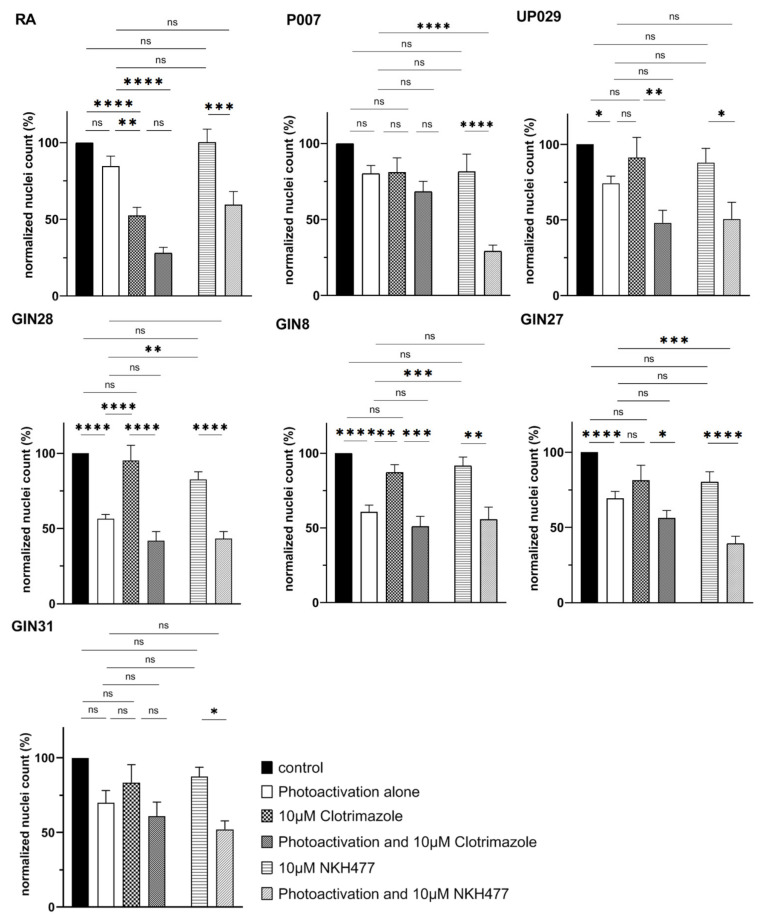
Synergistic effect of a low-dose NKH477 pre-treatment plus sub-lethal photoactivation (20 s) of TMRM-loaded GBM cells resulted in a significant decrease in the viability of some GBM cell lines (in UP007 and GIN27) compared to the controls or treatment with NKH477 alone or TMRM-PDT alone. No significant effect of clotrimazole after TMRM-PDT was observed. The number of independent experiments is: 5 for RA, 4 for UP007, UP029, GIN8, and GIN28 cells and 3 for GIN27 and GIN31 cells—all in duplicates. (ns) *p* > 0.05, (*) *p* < 0.033, (**) *p* < 0.002, (***) *p* < 0.0002, (****) *p* < 0.0001, alpha = 0.05. ns—not significant.

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
