# Peer review of "Feasibility of Photodynamic Therapy for Glioblastoma with the Mitochondria-Targeted Photosensitizer Tetramethylrhodamine Methyl Ester (TMRM)"

_biomedicines, 2021, doi:10.3390/biomedicines9101453_

Round 1

Reviewer 1 Report

Please clarify what is RA in 3.1.

Line 434-437 - how will this protocol tweaking work in an in vivo setting to reduce toxicity on non-glioblastoma cells? line 451 and 452 do speak to that but please elaborate further. 

Please include either data or description of intent from a future in vivo study since this will molecule will need to be validated in an in vivo setting. 

Author Response

1) Please clarify what is RA in 3.1.

Please note that the abbreviation «RA» was introduced in section 1. RA – rat astrocytes. We are therefore obliged to keep using it throughout the text.

2) Line 434-437 - how will this protocol tweaking work in an in vivo setting to reduce toxicity on non-glioblastoma cells? line 451 and 452 do speak to that but please elaborate further.

This has now been clarified as following (ln 436).

However, by carefully tweaking the protocol in terms of TMRM loading concentration and illumination times, we were able to achieve significant cytotoxic selectivity to GBM cells over RA (Figure 4). The first strategy is to use lower concentrations of TMRM, which might help to achieve a selective suppressant effect on GBM cells but not healthy cells. The ability of GBM mitochondria to accumulate more TMRM and retain it better matches with the generally known tendency of tumor cells to have hyperpolarized mitochondria [19,20]. The other approach is to further develop the protocol where photoactivation takes place after a delay, giving the cells time to dissipate the TMRM initially loaded into them. In our experiment this was demonstrated using 48 hours delay (Figure 6). We assume that the selective suppression of the GBM cells was due to the retention of the dye in their mitochondria because of the hyperpolarized MMP. This should result in much slower clearance of the dye from the GBM compared to the normal cells. From the clinical prospective this protocol would require a preloading of the tissue with TMRM prior to light application, possibly after a few hours. In fact, unless the molecules are actively retained in brain cells in vivo, they are usually very quickly washed away into the general circulation. Thus, in the living brain it might only take 1-2 hours for the healthy cells to release TMRM while the GBM cell could still have it concentrated in their mitochondria.

3)Please include either data or description of intent from a future in vivo study since this will molecule will need to be validated in an in vivo setting.

We have amended the CONCLUSIONS section as following:

Before TMRM can be suggested for clinical application, it will need to undergo further toxicological tests in vitro and in vivo. In view of its proposed topical application, some of the usual concerns, such as liver or kidney toxicity might be less problematic because the overall dose delivered into the brain will be fairly small, especially because only a periphery of the postoperative cavity needs to be impregnated with it. As the first step one could start by testing consequences of TMRM injections into the brain of experimental animals and then checking for the signs of pathology and inflammation.

We envision a treatment protocol for GBM where, following surgery, the walls of the cavity and ~ 2 cm of the surrounding parenchyma are infiltrated with TMRM and, after a delay that allows healthy brain cells to expel the photosensitizer, light is delivered directly into the brain parenchyma where disease infiltrations are suspected.

Reviewer 2 Report

In this manuscript by Vasilev et al, the authors show results of their studies on the potential of a photodynamic therapy (PDT) on glioblastoma (GBM) cells following selective uptake of the dye Tetramethylrhodamin methyl ester (TMRM), a dye often used to examin mitochondrial membrane potential. By treating either control primary rat astrocytes (RA) or different lines derived from GBM, they show that TMRM appears to accumulate in the mitochondria of both types of cells, but that PDT treatment causes rapid loss of mitochondrial membrane potential even in the control RA. However, further refinement of the concentration and light treatments identified a milder regimen that specifically affected the GBM but not RA cells, indicating this might be an effective therapy for patients. The authors use several assays for testing viability, including percentages of DAPI-stained nuclei, LDH assays and PrestoBlue staining, and they conclude with assays to demonstrate activation of caspase as a measure of apoptosis. The results support the authors claim that this technique holds promise as an option of glioblastoma treatment, but some issues should be considered by the authors as follows.

1) Line 206, the authors should be consistent with "TMRM-PDT" and not PDT-TMRM (or at least use only one version throughout, and "TMRM-PDT" makes the most sense).

2) Line 292, for the description of Figure 1b, more is needed than the phrase provided, e.g., "Mitochondira/nucleus fluorescence ratios decreased after 250 sec following light activation (assuming this is what is shown, which is not clear).

3) Lines 294-296 (Figure 1c legend), and text 302-304, the authors indicate from images in Figure 1c that TMRM is localized to the mitochondria, which then is dissipated after photoactivation, but localization to the mitochondria is difficult to assess in the images - while dissipation is evident in the "after' images, whether signal is localized to the mitochondria is not shown in the "before" images. Either show a higher magnification that indeed indicates mitochondrial localization, or add a mitochondria-specific label to show co-localization with the TMRM (the authors mention that mitochondrial localization was confirmed and shown to be stable, which they do not show, but here would be good to indicate that signal is lost specifically from the mitochondria, thereby indicating mitochondrial depolarization).

4) Figure 1c, which cells are being shown? This should at least be indicated in the legend.

5) Section 3.3, the authors show in the above figure (Fig. 3) that 300 nM TMRM causes significant loss of DAPI nuclei in RA cells, albeit with 40 seconds of PDT, but then in this figure (Fig 5) they show no effects of 300 nM for 30 seconds. Why did they change the duration of PDT, and did the 40 seconds show similar affects vs. those on UP007 and UP029 cells? This is confusing and needs to be addressed. Also, did they perform a viability test to correlate percentages of DAPI-stained nuclei with cytotoxicity of the treatment, e.g., MTT assay or even the LDH assay used later? It would be good to show a correlation for the loss of viability with the nuclear staining test.

6) Lines 356-357, and Figure 6, the authors loaded RA and GBM cells with TMRM and then assessed % DAPI cells as an indicator of TMRM accumulation in the mitochondria, but this is not actually shown in the figure - just because they show increased loss of cells via loss of DAPI-stained nuclei, they cannot state that this indicates preferential accumulation of TMRM in GBM cells, as compared to the RA cells.

7) Figure 8a, the figure is confusing when reading the figure legend - the data is indicating Cell Viability (presumably with LDH activity levels), but the legend indicates mitochondrial/nucleus fluorescence, which is not shown in the figure. Moreover, this is also the case with figure 8b, and more importantly, is inconsistent with the decreased DAPI staining shown for UP007 and UP029 cells in Figure 6. The data and/or explanations need to be edited or better explained. Furthermore, why would PDT on the TMRM-loaded cells not show loss of viability? Perhaps this reviewer is confused by the descriptions.

8) Figure 9, did the authors examine TMRM-PDT on RA cells for Presto-Blue to assess cell viability, as a control vs. the GBM cells? This would be good to show.

9) In Methods (lines 141-153) the authors describe use of the MTT assay, but none of the figures or results mention data generated with this assay – was this the assay used in Figure 8, which indicates in the legend the graphs show mitochondria/nucleus fluorescence, and yet the data is given as cell viability (%)? This needs to be addressed.

Author Response

Thank you for your time and effort and many valuable suggestions.

Specific answers are below.

1) Line 206, the authors should be consistent with "TMRM-PDT" and not PDT-TMRM (or at least use only one version throughout, and "TMRM-PDT" makes the most sense).

Corrected

2) Line 292, for the description of Figure 1b, more is needed than the phrase provided, e.g., "Mitochondria/nucleus fluorescence ratios decreased after 250 sec following light activation (assuming this is what is shown, which is not clear).

Corrected, as suggested.

3) Lines 294-296 (Figure 1c legend), and text 302-304, the authors indicate from images in Figure 1c that TMRM is localized to the mitochondria, which then is dissipated after photoactivation, but localization to the mitochondria is difficult to assess in the images - while dissipation is evident in the "after' images, whether signal is localized to the mitochondria is not shown in the "before" images. Either show a higher magnification that indeed indicates mitochondrial localization, or add a mitochondria-specific label to show co-localization with the TMRM (the authors mention that mitochondrial localization was confirmed and shown to be stable, which they do not show, but here would be good to indicate that signal is lost specifically from the mitochondria, thereby indicating mitochondrial depolarization).

We have included images taken at higher magnification on a confocal microscope (1d).The legend now is:

(d) An example of a typical response to TMRM-PDT of a GBM GIN8 cell. Initially (left image) TMRM is localized exclu-sively to the mitochondria (red arrows) while nucleus (blue arrow) is almost completely devoid of the staining. After photoactivation (right image) TMRM leaves the mitochondria and the contrast between mitochondria and nucleus is lost, hence the ratio mitochondria/nucleus drops). For clarify images ware taken at high magnification using LSM780 confocal microscope.

4) Figure 1c, which cells are being shown? This should at least be indicated in the legend.

Corrected, description in the text now:

Typical examples TMRM distribution in GIN8 cell line before and after photoactivation are shown in Figures 1c and 1d.

5) Section 3.3, the authors show in the above figure (Fig. 3) that 300 nM TMRM causes significant loss of DAPI nuclei in RA cells, albeit with 40 seconds of PDT, but then in this figure (Fig 5) they show no effects of 300 nM for 30 seconds. Why did they change the duration of PDT, and did the 40 seconds show similar affects vs. those on UP007 and UP029 cells? This is confusing and needs to be addressed.

Thank you for pointing it out. In this case we were trying to apply a milder photostimulation regime in order to not completely lose all the cells with 800 nM TMRM loading, hence duration was shortened by 10 sec. For reasons we do not understand the effect of time does not seem to be very linear, possibly this depends on reaching some critical threshold which may be different for difference cell types. We will address this in our next experiments.

However, we agree that the lack of effect on RA in this series looks inconsistent with Figure 3. In order to make this less confusing we are not discussing other concentrations but highest (800) and lowest (100) in the text. There is a clear dependence of the effect on the concentration of TMRM. What was also lacking in the description was a mention of significantly stronger overall impact on the GBM lines with 800 nM under these conditions.

We have removed the statement of the effect being preferential to GBM.

Text now reads:

3.5. Effect of TMRM is concentration-dependent

In order to better demonstrate dependence of TMRM effect on its concentration we applied different concentrations while using slightly shortened the duration of illumination (30 sec) in order to preserve at least some cells in stimulated wells, which was important for counting purposes. The number of DAPI positive nuclei was not affected with TMRM loading at 100nM while 800nM significantly affected all tested cell types (Figure 5). In fact, the effect on both GBM cell lines was much greater than on RA (p<0.001 in both cases).Legend now reads

Figure 5. Dependence of TMRM-PDT (0 nM, 100nM and 800nM for 30 seconds; "1.4mW/" 〖"mm" 〗^"2" ) on TMRM concentration. n is the total number of repeats from 4 independent experiments. (****) p value <0.001. Differences between the effect of 800 nM on RA and either of the GBM cell lines were highly significant (p<0.01 in either case).

Also, did they perform a viability test to correlate percentages of DAPI-stained nuclei with cytotoxicity of the treatment, e.g., MTT assay or even the LDH assay used later? It would be good to show a correlation for the loss of viability with the nuclear staining test.

It was not possible to combine these measurements, although we agree, it would be useful. Both MTT and LDH are “an mass” assays and they require plenty of cells in each well, otherwise the readings are too low. For the most part where we stained nuclei, as indicated in methods, we seeded a small number of cells in each well (around 300) so that when we counted nuclei we could be sure that all cells were evenly illuminated. To test TMRM dark toxicity where we used LDH, illumination was not required and we could plate more cells so that LDH assay could be performed.

6) Lines 356-357, and Figure 6, the authors loaded RA and GBM cells with TMRM and then assessed % DAPI cells as an indicator of TMRM accumulation in the mitochondria, but this is not actually shown in the figure - just because they show increased loss of cells via loss of DAPI-stained nuclei, they cannot state that this indicates preferential accumulation of TMRM in GBM cells, as compared to the RA cells.

This is a misunderstanding, we did not state that % of DAPI nuclei is an indicator of TMRM accumulation. DAPI only tells us how many cells survive (or are lost). We assume that the impact should be dependent on the degree of TMRM accumulation but we also demonstrate that this is not an absolute rule since GIN27, which initially had lower TMRM loading (Fig 1) nevertheless was not immune to the PDT effect (Fig 3). Accumulation of TMRM in GBM cells is also shown in supplementary figure S2

7) Figure 8a, the figure is confusing when reading the figure legend - the data is indicating Cell Viability (presumably with LDH activity levels), but the legend indicates mitochondrial/nucleus fluorescence, which is not shown in the figure. Moreover, this is also the case with figure 8b, and more importantly, is inconsistent with the decreased DAPI staining shown for UP007 and UP029 cells in Figure 6. The data and/or explanations need to be edited or better explained. Furthermore, why would PDT on the TMRM-loaded cells not show loss of viability? Perhaps this reviewer is confused by the descriptions.

The legend was confusing and we have changed it, hopefully it is clearer now.

Figure 8. Further control experiments to assess dark toxicity of TMRM and the effect of green light. (a) Prolonged incubation with TMRM for 48 hours with no photoactivation had minimal effect on GBM or RA cell density as measured using MTT assay. (b) When applied to unloaded cells, green light (1.4mW/2 ) on its own, had no effect on the survival of GBM cell lines or RA (PrestoBlue assay). n: is the total number of repeats for each condition from 3 independent experiments (6 for RA in panel A.). (*) p <0.05, (**) p<0.01, (***) p<0.001. Data presented as the mean ± SEM.

8) Figure 9, did the authors examine TMRM-PDT on RA cells for Presto-Blue to assess cell viability, as a control vs. the GBM cells? This would be good to show.

Thank you for this suggestion, we agree that this could be interesting but the purpose of this experiment was rather to see if the few surviving GBM cells were basically unaffected by PDT or could quickly restore their metabolism which was not the case. We will consider this experiment in our future work. The present study is only a pilot and we clearly have a lot more work to address many of the interesting questions which have been raised.

9) In Methods (lines 141-153) the authors describe use of the MTT assay, but none of the figures or results mention data generated with this assay – was this the assay used in Figure 8, which indicates in the legend the graphs show mitochondria/nucleus fluorescence, and yet the data is given as cell viability (%)? This needs to be addressed.

Thank you, we have corrected this issue in the legend to Figure 8 (see above).

Reviewer 3 Report

Biomedicines-1385115

The MS written by Vasilev and Sofi et al. is dealing with the feasibility of photodynamic therapy for glioblastoma using TMRM, a rhodamine derivative, a compound showing mitochondrial membrane potential (MMP) dependent uptake into mitochondria. The compound is well-known in mitochondrial research because it is a fluorophore and widely used for MMP determination. The idea is original, use the TMRM as a photosensitizer, it will be taken up by mitochondria and low intensity green light will induce mitochondrial damage. The damage is indicated by the depolarization of MMP and by the loss of mitochondrial selectivity of TMRM’s localization. Authors detected selective killing of glioblastoma multiforme cells but not that of rat astrocytes.

The paper is well written, the results are solid, but some issues should be clarified.

Major points:

Please discuss the possible neuronal damage as a consequence of phototherapy.

Authors put emphasis to the description of methods; however for the referee this part of the MS is still a bit chaotic.

Comments and questions

Introduction

39-40 reference required

46 ketogenic diet – would it be worth to mention?

Methods

137-138 „where the amount of LDH in a sample is proportional to the amount of red product produced by the assay reactions.” What is the product?” Is the LDH enzyme activity measured?

155-158 What is the basis of this viability measurement? How can it be repeated? Washing the cell culture, or spontaneous decomposition?

161-163 What is the wavelength used for photoactivation? Is it the same as the excitation wavelength used for fluorimetric measurement of TMRM?

163-164 absorbance required for reference was measured at 600nm.” The absorbance of what?

Comment affecting the whole MS: It would be helpful to describe the wavelength of photoactivation, the wavelength of excitation and the wavelength of emission for TMRM detection.

162-164 „Briefly, 10uL of PrestoBlue reagent was added to each well and incubated for 20 min. Fluorescence was measured at 560nM, and absorbance required for reference was measured at 600nm” The excitation wavelength is shorter than that of emission. If we talk about fluorescent measurement using the term absorbance I think is misleading.

178 „baseline images were obtained as a sequence of 6 images, one every 10 seconds, for a total of one minute” How were the baseline images detected? Excitation/emission wavelengths, bandwidth or the type of filter used are necessary.

193-194 „To ensure that all cells were evenly illuminated, we used a special plating technique which ensured that they were localized in the centre of the well” Question: did authors use any specific type of plate for fluorescence measurements?

Results

347-350. How is it possible to quantify the higher mitochondrial membrane potential detected in tumour cells?

Generally speaking it would be desirable to establish a zero membrane potential condition using uncoupler and a maximal MMP using oligomycin,

360 „We controlled for detrimental effects of TMRM without light application (dark toxicity) for RA and all GBM lines.” However the title of Fig.: Control tests for TMRM toxicity and the effect of light. In this form the title of Fig.7 is misleading, because cells were not illuminated.

Fig.9 Please consider changing the title of y axis to „% of control”. Also in the second graph of Fig.9, below the first column the tick label is in Russian.

Fig. 10 The control columns should also have error bars.

Author Response

Major points:

Please discuss the possible neuronal damage as a consequence of phototherapy.

In this work we did not study potential impact of PDT on neurons, but this is a particularly difficult task in vitro. Essentially all work on cultured rodent neurons employs cells from embryos which have a very metabolism to the mature neurons in the brain in vivo. We do not believe that such cultures would be a suitable model for this type of work. Instead it may, perhaps it will be better to test whether TMRM strongly affects neurons in rodent studies in vivo, as we hope to do in future studies.

Authors put emphasis to the description of methods; however for the referee this part of the MS is still a bit chaotic.

The reviewer did not specify what part and we cannot specifically address this comment.

Comments and questions

Introduction

39-40 reference required

Corrected

46 ketogenic diet – would it be worth to mention?

We considered this suggestion but after some discussions could not fit it into the text without it looking artificial.

Methods

137-138 „where the amount of LDH in a sample is proportional to the amount of red product produced by the assay reactions.” What is the product?” Is the LDH enzyme activity measured?

We have added this explanation:

This is a colorimetric assay where the amount of LDH in a sample is proportional to the amount of red formazan product produced by the NADH, generated by the LDH-mediated conversion of L-lactate.

155-158 What is the basis of this viability measurement? How can it be repeated? Washing the cell culture, or spontaneous decomposition?

The reagent, according to the manufacturer is “cell permeable resazurin-based solution that functions as a cell viability indicator by using the reducing power of living cells to quantitatively measure the proliferation of cells.” It can be used to assess the intensity of fluorescence as a proxy of cell viability, washed off and applied again and again.

161-163 What is the wavelength used for photoactivation? Is it the same as the excitation wavelength used for fluorimetric measurement of TMRM?

This obviously relates to the Presto Blue reagent. Yes, excitation was at ~ 540-550 nm but please note, that the plate reader uses only very short exposures (0.1-0.2 sec) to take images.

We changed the text to explain:

Fluorescence was measured at 560nM, thus reflecting the amount of the fluorescent product of conversion of the reagent.

163-164 absorbance required for reference was measured at 600nm.” The absorbance of what?

We removed this line, it was confusing the matter but with this reagent one can assess the total quantity of the reagent using this parameter.

Comment affecting the whole MS: It would be helpful to describe the wavelength of photoactivation, the wavelength of excitation and the wavelength of emission for TMRM detection.

We did not use a laser which precludes mention of a specific wavelength. The band-pass filters which are usually used to visualise rhodamine-like stains and which give green light from a white light lamp were used.

We added this line in section 2.7.1:

Green light was used using standard filter blocks of Leica microscopes with a pass-band ~515-560 nm.

162-164 „Briefly, 10uL of PrestoBlue reagent was added to each well and incubated for 20 min. Fluorescence was measured at 560nM, and absorbance required for reference was measured at 600nm” The excitation wavelength is shorter than that of emission. If we talk about fluorescent measurement using the term absorbance I think is misleading.

CORRECTED – see above.

178 „baseline images were obtained as a sequence of 6 images, one every 10 seconds, for a total of one minute” How were the baseline images detected? Excitation/emission wavelengths, bandwidth or the type of filter used are necessary.

Thank you for pointing this out. This was to indicate that they were simply regularly taken photographs without a period on photostimulation. All images were taken using the standard rhodamine filter blocks.

The text now reads:

Before photoactivation of TMRM, baseline images using standard rhodamine filter block of Leica microscopes (excitation 515-560 nm, emission - high pass filter 580 nm) were obtained as a sequence of 6 images, one every 10 seconds, for a total of one mi-nute. This was followed by constant illumination with same green light for 30 seconds (photoactivation) followed by a series of 20 images every 10 seconds, for a total of 3 minutes.

193-194 „To ensure that all cells were evenly illuminated, we used a special plating technique which ensured that they were localized in the centre of the well” Question: did authors use any specific type of plate for fluorescence measurements?

No, these were standard 96 wells. However, we noticed that in regular settings many GBM cells attached to the corners and even walls of the wells. Therefore we had to make sure that all cells were plated on the bottom.

Results

347-350. How is it possible to quantify the higher mitochondrial membrane potential detected in tumour cells?

This is based solely on the brightness of TMRM, which is a standard approach.

Generally speaking it would be desirable to establish a zero membrane potential condition using uncoupler and a maximal MMP using oligomycin,

Agreed, but in these experiments we did not do that. We will take this suggestion onboard for future work, thank you.

360 „We controlled for detrimental effects of TMRM without light application (dark toxicity) for RA and all GBM lines.” However the title of Fig.: Control tests for TMRM toxicity and the effect of light. In this form the title of Fig.7 is misleading, because cells were not illuminated.

Title of the legend now says “TMRM dark toxicity”.

Fig.9 Please consider changing the title of y axis to „% of control”. Also in the second graph of Fig.9, below the first column the tick label is in Russian.

CORRECTED

Fig. 10 The control columns should also have error bars.

This depends on how the control is calculated. In our case we took all measurements in relation of their own controls which therefore were always 100%, hence there is no variability and no error marks.

Reviewer 4 Report

The manuscript describes the cytotoxicity of TMRM-PDT on GBM cells. Several conditions including the concentrations of TMRM, irradiation intensity, selection of wavelength, irradiation time, and combination of reduced irradiation intensity with inducer/inhibitor were examined to maximize the efficacy of the TMRM-PDT treatment. The authors also proposed a concept of delayed photoactivation of TMRM for the development of therapeutic strategy based on the different TMRM mitochondria existing period between normal and cancer cells. This is an intriguing study and provides valuable information. The manuscript is organized prepared, and the results were logically presented. The data fully supported the authors’ claim. Some issues could be responded before publishing.

  1. English editing service is required throughout the manuscript.
  2. Although the functions of NKH-477 and TMRM could be found in the section of “discussion,” the author could briefly introduce these agents in the section of “Introduction.”
  3. A typo in line 272.
  4. In “Fig7,” the significance should be labeled.
  5. In “Fig 8,” the descriptions of legends did not match with the figure.
  6. The intracellular ROS levels can be measured after TMRM-PDT treatment since the production of ROS is one of the cytotoxic mechanisms of PDT. This could give some hints on the heterogenous responses of GBM cells.

Author Response

English editing service is required throughout the manuscript.

Please note that several authors are native English speakers, they read the manuscript.

Although the functions of NKH-477 and TMRM could be found in the section of “discussion,” the author could briefly introduce these agents in the section of “Introduction.”

Added, as requested.

In line 97 now:

TMRM, which is a rhodamine derivative driven into the mitochondria by its negative membrane potential, has been in routine laboratory use as a research reagent but never tested as a potential therapeutics.

A typo in line 272.

corrected

In “Fig7,” the significance should be labeled.

corrected

In “Fig 8,” the descriptions of legends did not match with the figure.

corrected

The intracellular ROS levels can be measured after TMRM-PDT treatment since the production of ROS is one of the cytotoxic mechanisms of PDT. This could give some hints on the heterogenous responses of GBM cells.

We have tried measure Mitochondrial ROS with dye MitoSOX , but unfortunately did not get clear results (please note, these in this case several fluorescent molecules interfere with each other). However, other methods can be used in our future work.

Round 2

Reviewer 2 Report

The authors have addressed the concerns of this reviewer, and the manuscript is both clearer and improved in quality. There are some minor typos or grammatical errors, so suggest a careful review of the text.